# Incidence and determinants of Implanon discontinuation: Findings from a prospective cohort study in three health zones in Kinshasa, DRC

**P. Z. Akilimali**[1]*, **Hernandez J.**[2], **Anglewicz P.**[3], **Kayembe K. P.**[1], **Bertrand J.**[2]

**1** Kinshasa School of Public Health, University of Kinshasa, Kinshasa, the Democratic Republic of the Congo, **2** School of Public Health and Tropical Medicine, Tulane University, New Orleans, Louisiana, United States of America, **3** Department of Population, Family and Reproductive Health, Johns Hopkins Bloomberg School of Public Health, Baltimore, Maryland, United States of America

* pierretulanefp@gmail.com

## Abstract

**Data Availability Statement:** All relevant data are within the manuscript and its Supporting Information files.

### Background

Kinshasa is Africa's third largest city and one of the continent's most rapidly growing urban areas. PMA2020 data showed that Kinshasa has a modern contraceptive prevalence of 26.5% among married women in 2018. In Kinshasa's method mix, the contraceptive implant recently became the dominant method among contraceptive users married and in union. This study provides insight into patterns of implant use in a high-fertility setting by evaluating the 24-month continuation rate for Implanon NXT and identifying the characteristics associated with discontinuation.

### Methodology

This community-based, prospective cohort study followed 531 Implanon users aged 18–49 years at 6, 12 and 24 months. The following information was collected: socio-demographic characteristics, Method Information Index (MII) and contraceptive history. The main outcome variable for this study was implant discontinuation. The incidence rate of discontinuation is presented as events per 1000 person/months (p-m), from the date of enrolment. The Cox proportional hazards modelling was used to measure predictors of discontinuation.

### Results

A total of 9158.13 p-m were available for analysis, with an overall incidence rate of 9.06 (95% CI: 9.04–9.08) removals per 1000 p-m. Of nine possible co-variates tested, the likelihood of discontinuation was higher among women who lived in military camps, had less than three children, never used injectables or implants in the past, had experienced heavy/prolonged bleeding, and whose MII score was less than 3.

**Funding:** APZ: received grant from NIH Fogarty International Center, (grant number #D43TW009340) for his PostDOC program JB received funding to implement the distribution of contraceptive methods at the community level from the Bill and Melinda Gates Foundation (OPP1128892), and UNFPA (Project ID COD04CSF). The funders had no role in study design, data collection and analysis, decision to publish, or preparation of the manuscript.

**Competing interests:** I have read the journal's policy and the authors of this manuscript have the following competing interests: The authors declare no conflict of interest. The organizations cited above supported the study financially but did not have a role in the design of the study, analysis of the data, or final interpretation of findings.

## Conclusion

In addition to four client characteristics that predicted discontinuation, we identified one programmatic factor: quality of counseling as measured by the Method Information Index. Community providers in similar contexts should pay more attention to clients having less than three children, new adopters, and to clients living military camps as underserved population, where clients have less access to health facilities. More targeted counselling and follow-up is needed, especially on bleeding patterns.

## Background

The Democratic Republic of the Congo (DRC) is Africa's fourth most-populous and one of the region's fastest growing countries [1]. DRC has one of the highest fertility rates in the world: the most recent Demographic and Health Survey (DHS) estimated a country-level total fertility rate of 6.6, a slight increase since the 6.3 TFR estimated with the 2007 DHS [2, 3]. Contraceptive use is low in DRC: the modern contraceptive prevalence rate (MCPR) among women aged 15–49 years married or living in union is estimated at 7.8% for the country as a whole [3].

Kinshasa, the capital of the DRC, is Africa's third largest city (after Lagos and Cairo) and one of the continent's most rapidly growing urban areas [2,4]. PMA2020 data showed that Kinshasa had a modern contraceptive prevalence rate (MCPR) of 26.5% among women married or in union in 2018 [5], and 64.1% of last births were unintended [5]. Although the MCPR is similar to other major cities in francophone sub-Saharan Africa (SSA), it remains low in comparison to the capital cities of nearby Anglophone countries, such as Nairobi, 58.3% [6], Harare, 58.2%[7], Lusaka, 54.7%[8], and Kampala, 40.2%[9]. As of the most recent PMA2020 survey in Kinshasa, the implant surpassed condoms as the dominant method among FP (family planning) users married and living union [5].

In the DRC, only trained physicians and nurses are typically authorized to provide injections and insert implants. In late 2016 Tulane University began a pilot study in Kinshasa (in three health zones) of an innovative approach to test the acceptability and feasibility of having nursing students insert Implanon NXT at the community level. Women interested in a contraceptive method participated in a series of mini-campaigns at a community location (not in a fixed clinic); the nursing students provided a range of five methods, including Implanon NXT insertion to interested clients.

As part of their practical training, the students provided FP services at the community level: at outreach events (mini-campaigns) set up in a central location (e.g., near a referral health facility, beside a market), door-to-door, or from their own homes. Nursing students offered a range of five contraceptive methods: Implanon NXT, DMPA-SC, oral pills, condoms, and Cycle beads. Prospective clients often preferred these community services to fixed facilities in their neighbourhoods, because they were less expensive and required less waiting time.

Although Implanon NXT is generally well tolerated, previous studies have indicated that a proportion of women will discontinue its use because of unacceptable side effects [10, 11]. Another study showed that implant, injectable, and IUD users experienced more disturbances in bleeding patterns than pills users [12]. The acceptability of side effects of a given contraceptive method influences its compliance rates. It was observed that the women having frequent or prolonged bleeding had discontinued the contraceptive method more often as compared to those having delayed bleeding episodes or oligomenorrhoea [12].

Providing women with a choice of contraceptive methods and high-quality counselling are essential components of rights-based family planning (FP) [13]. Although Implanon is gaining

in popularity among contraceptive users living in union in Kinshasa [5], virtually nothing is known about continuation rates, reasons for discontinuation, or the quality of services available for removal. Implant discontinuation, while being an important outcome of the quality of FP services [14, 15], has never been studied in the DRC context.

Additionally, studies reporting the use and continuation rates of Implanon are scarce, particularly those conducted at the community level (outside a fixed clinic). We did not find any studies that have quantitatively documented discontinuation rate among user in a community setting. This research provides insights into patterns of implant use in a high-fertility country by evaluating the 24-month continuation rate for Implanon and identifying the characteristics associated with continuation of this method at 24 months. This article also highlights some of the reasons that clients chose to have the method removed. The findings of this research are useful in documenting continuation rates for Implanon provided in a community setting in Africa, as well as identifying factors related to discontinuation.

## Methodology

### Population and design

Between November 2016 and January 2019, we conducted a cohort study of 531 women who chose to use Implanon NXT. The study took place in three selected health zones of Kinshasa: Masina I, Ndjili and Kokolo. The selection of zones took into account the proximity of the training institutes in which the students were trained. In addition, for logistical efficiency, we needed to limit that number of health zones to be covered and selected three with different characteristics to capture different types of clients. Kokolo health zone consists of compounds that house military personnel and their families. Military compounds differs from other health zones in that they are a closed encampment within the city with their own health system, though often underfunded and affording less access to family planning than in the non-military population.

Nursing students from local nursing schools received training in contraceptive counseling and provision, including insertion of Implanon NXT. They provided FP services at the community level as part of their practical training. Between October 2016 and January 2017, a total of 1,010 prospective clients came to "four campaign days" (community outreach events) at which they received counselling on a range of methods; Out of the 702 women who received Implanon NXT, 531 women aged 18 to 49 years old agreed to participate in the study and gave their informed consent to be interviewed at 6,12 and 24 months. The research team obtained participants' addresses and phone numbers at baseline, then made phone calls to locate these acceptors and schedule in-person follow-up interviews. If needed, two community agents in each health zone were tasked with contacting acceptors at the addresses they had provided.

An experienced team of 2 supervisors and 12 data collectors were trained for three days before data collection at baseline and two days before the subsequent rounds of data collection. The interviewers used a structured questionnaire to collect data at baseline, 6, 12, and 24 months. The questionnaire was translated into Lingala, the most commonly spoken language in Kinshasa, during training sessions. The data collection tools were pre-tested and checked for consistency, then modified as deemed necessary. Interviewers collected the data on smartphones equipped with the SurveyCTO application. Supervisors monitored the data collection and checked the data for completeness before submission to a secure server. They also held regular meetings with the interviewers to ensure data quality.

For the follow-up interviews, respondents were asked to appear at a central community location. Those who did not were called or visited by the two community agents, based on information collected at baseline. Women were considered lost to follow-up if they could not

be located at the address or at the phone number(s) given, after 3 attempts to contact them at 6, 12 or 24 months. Participants received U.S. $5.00 as reimbursement for transportation costs per follow-up visit.

## Measurement of outcomes and other variables of interest

We collected socio-demographic characteristics (age, gender, marital status, education, socio economic status [SES], number of living children, and residence (military camps vs non-military camps), as well as the Method Information Index (MII) and contraceptive history. Data collectors recorded the date of Implanon insertion. At each follow-up visit acceptors were asked about side effects (especially if they experienced heavy bleeding), if the Implanon was still in place, and (if not), the reasons for removal of the implant and the date of removal.

## Outcome definitions and measures

Participants were closely monitored at 6, 12, and 24 months post-insertion. The main outcome variable for this study was method discontinuation. Data collectors verified the presence of implant in the woman's left arm at each interview visit. Time until removal was treated as a continuous variable, measured in months, with a maximum allowable time of 24 months. The event of interest was defined as discontinuation of Implanon use.

**Covariates.**   The MII (Method Information Index) is a proxy measure for quality of counselling. It is calculated by summing the binary responses of clients to the following 3 questions: "During your visit today, were you told about other methods of family planning that you could use?"; "Were you told about side effects or problems that you might have with (your chosen) method?"; and "Were you told what to do if you experienced side effects or problems?" The index, ranging from 0 to 3, was used as an ordinal variable as well as a binary variable (3 or less than 3) in the analyses. The MMI is used when direct observation of the client provider interaction is not possible; it captures a woman's recall and understanding of the information exchanged at the time of adoption, in addition to whether the exchange occurred [16].

The household wealth index was constructed based on principal component analysis [17] to create an index from a set of household assets (e.g. radio, tape recorder, television set, bicycle, flashlight), housing conditions (roof material, number of rooms, wall type, windows, availability and type of latrine), and ownership of livestock. The study participants were classified according to the wealth index score, into three groups (high, medium, low). The Cronbach's alpha was 0.76.

We recoded respondent's age as 18–24 and 25–49 years. In terms of education, women with less than a high school education were classified as "less than secondary school" and "completed secondary school.". Residence fell into one of two categories: military area or non-military area. We recoded marital status as "married/living in union" or "not married." And we classified number of children as less than 3 children versus 3 children and more.

The responses to the question on experience of heavy/prolonged bleeding was binary: yes or no. Women who needed to change their tampon or pad after less than 2 hours during more than 7 days were considered as having prolonged or heavy bleeding. Because no respondents had ever used the IUD, we classified contraceptive history into two categories: ever used injectable or implant and never used either of these methods.

## Analytic methods

Interviewers collected data using a mobile phone (SurveyCTO)., then submitted it to a cloud server. Do-file checking and cleaning were updated daily, and errors discovered were corrected in the field. We completed a final data cleaning prior to data analysis.

All analyses were carried out using Stata version 15. For continuous variables, means and standard deviation (SD) were calculated; for categorical data, proportions and their respective 95% confidence intervals were calculated. We used the chi-square test and Fisher´s exact test when appropriate.

The incidence rate of discontinuation is presented as events per 1,000 person/months (p-m), from the date of enrolment. The Kaplan-Meier curve was used to determine the probability of discontinuation as a function of time of inclusion in the cohort. The log-rank test was employed to compare survival curves based on determinants. The Cox proportional hazards modeling allowed us to identify predictors of discontinuation from the date of insertion of the device (Implanon) to the end point, which was set at 24 months. The following variables were considered for inclusion to the Cox regression model: age, education, residence, marital status, wealth level, number of living children, contraceptive history, experience of heavy bleeding, and Method Information Index. This model provided the adjusted hazard ratios (AHR) and 95% confidence intervals (CI). The proportionality test based on Schoenfeld residuals allowed verification of compliance with the assumption of proportionality of risks. Variance-inflation factors (VIF) were estimated to assess multicollinearity. All tests were two-sided, and the level of significance was set at p <0.05.

### Ethical review

This study received IRB approval from Tulane University (16 –#911338) and the Kinshasa School of Public Health (ESP/CEI/071/ 2016 and ESP/CEI/125/ 2018).

## Results

### Sample description

Of 531 women enrolled in the study, 116 (22%) were lost to follow-up (LTFU), resulting in 415 cases for inclusion in the analysis (see Fig 1). Those LTFU were similar to those retained in the analysis in terms of education, matrimonial status, residence, number of living children, contraceptive history, MMI, and the experience of heavy bleeding. However, the women who

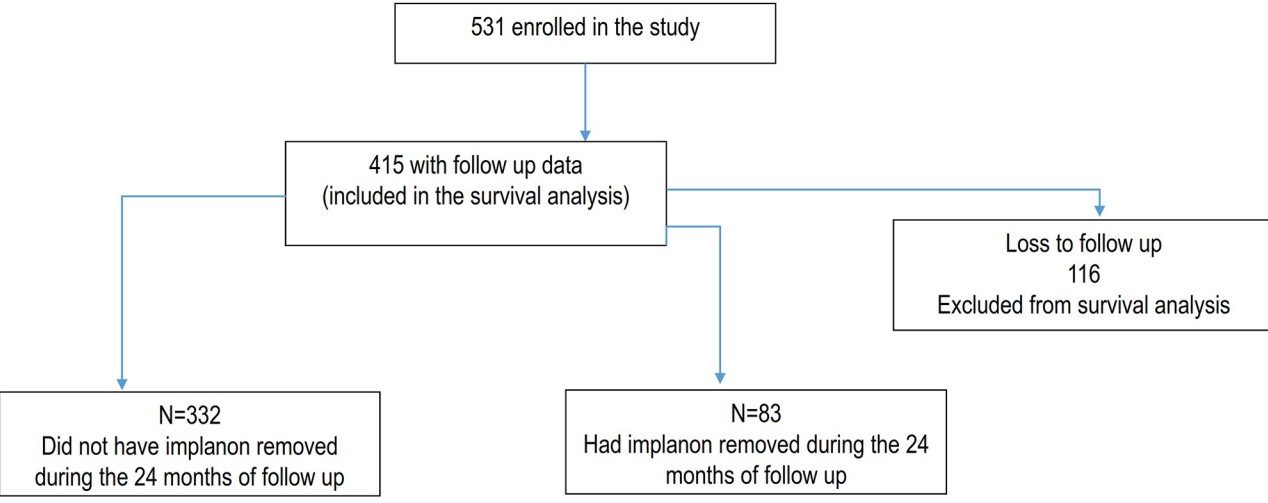

**Fig 1. Flow diagram showing the way follow-up of participants was done from the insertion end point.**

**Table 1. Sociodemographic characteristics and past contraceptive use of women LTFU and women included in the analysis.**

| | Lost to Follow-Up | | Included in the analysis | | p |
|---|---|---|---|---|---|
| | **n(116)** | **%** | **n(415)** | **%** | |
| Age: mean (SD) | 25.37 (5.22) | | 27.96 (6.39) | | < 0.001 |
| Instruction level | | | | | 0.374 |
| • Less than secondary school | 27 | 23.3 | 81 | 20.0 | |
| • Completed secondary school | 89 | 76.7 | 334 | 80.0 | |
| Matrimonial Status | | | | | 0.070 |
| • Not married | 103 | 88.8 | 339 | 82.0 | |
| • Married/partnered | 13 | 11.2 | 76 | 18.0 | |
| Residence | | | | | 0.653 |
| • Military area | 24 | 20.7 | 94 | 23.0 | |
| • Non military area | 92 | 79.3 | 321 | 77.0 | |
| SES | | | | | 0.015 |
| • Lowest third | 51 | 44.0 | 126 | 30.4 | |
| • Middle third | 36 | 31.0 | 141 | 34.0 | |
| • Highest third | 29 | 25.0 | 148 | 35.6 | |
| Number of living children | | | | | 0.303 |
| • ≤ 2 | 52 | 44.8 | 164 | 40.0 | |
| • 3 or more | 64 | 55.2 | 251 | 60.0 | |
| Contraceptive history | | | | | 0.487 |
| • Ever used injectable or implant | 10 | 8.6 | 45 | 11.0 | |
| • Never used injectable or implant | 106 | 91.4 | 370 | 89.0 | |
| Experienced heavy/ prolonged bleeding * | | | | | 0.174 |
| • Yes | 9* | 21.4 | 131 | 31.57 | |
| • No | 33* | 78.6 | 284 | 68.43 | |
| Method Information Index | | | | | 0.798 |
| • < 3 | 51 | 44.0 | 188 | 45.3 | |
| • = 3 | 65 | 56.0 | 227 | 54.7 | |

*Data collected before lost to follow up

were LTFU were younger (25 vs 28 years; p < 0.001) and lower on the wealth index (44% vs 30%, p = 0.015) than those remaining in the analysis. (Table 1).

## Profile of respondents at baseline

Among the 415 women retained in the analysis, the mean age was 28 years; 82.0% were unmarried; 60% had at least 3 children. Most (80.0%) had graduated from secondary school, and 23.0% lived in a military camp. In terms of family planning, 59% had never used an injectable or implant in the past; 55% reported receiving information about all 3 items in the Method Information Index; and a third (32%) reported experiencing heavy bleeding (Table 1).

## Evolution of discontinuation

From 415 women retained in the analysis, 83 (20%: 95% CI: 16.2–23.8) had removed the Implanon NXT during the 24-month study period. From the date of insertion, the removal rate was 5.5% (by 6 months), 8.4% (by 12 months), 10.1% (by 18 months) and 20.0% (by 24 months) (S1 Table).

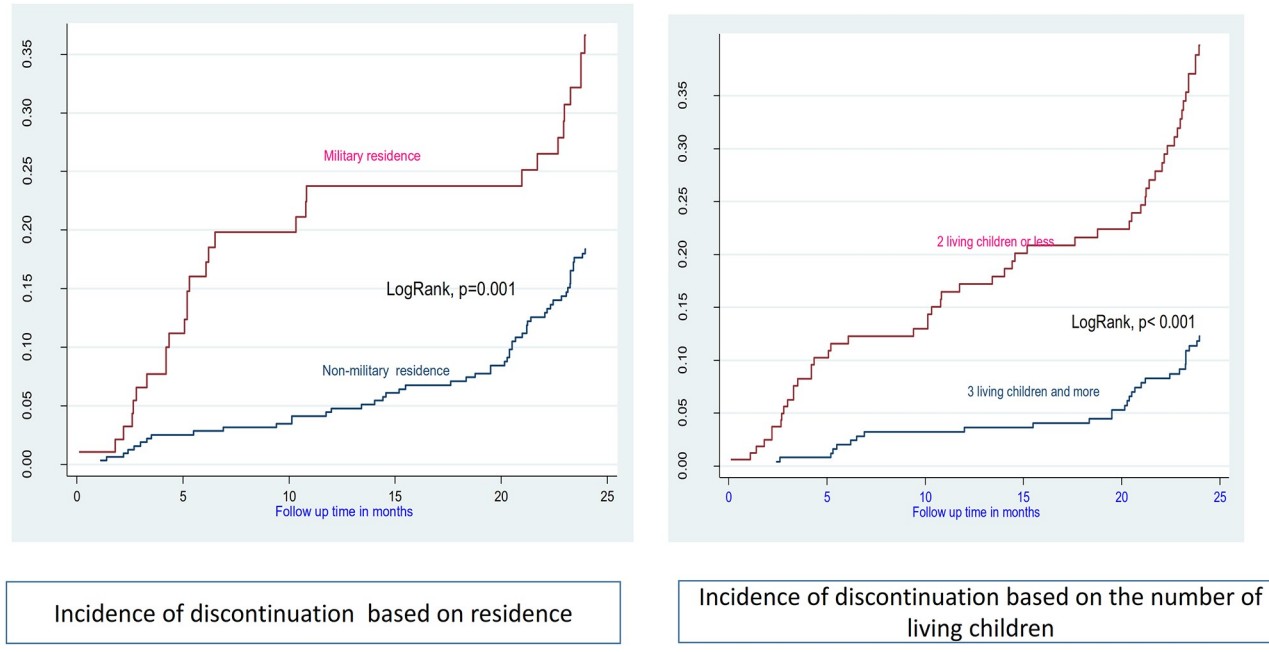

Incidence of discontinuation based on residence

Incidence of discontinuation based on the number of living children

**Fig 2. Incidence of discontinuation based on residence and parity.**

## Predictors of discontinuation in multivariate analysis

A total of 9158.13 p-m were available for analysis, with an overall incidence rate of 9.06 (95% CI: 9.04–9.08) removals per 1000 p-m (Figs 2 and 3).

We tested nine variables as possible co-variates of discontinuation. The likelihood of discontinuation was higher among women who lived in the military camps, who had less than three children, who never used injectables or implants in the past, who had experienced heavy/prolonged bleeding, and who score less than 3 on the MII (Table 2).

## Reasons for early discontinuation

The majority of discontinuers did so because of side effects (72.3%). The most commonly reported side effect was heavy bleeding (30.0%). Also, nearly one discontinuer out of five mentioned opposition of her partner/husband. Opposition of partner was cited more among discontinuers who experienced heavy bleeding than those who did not experience it (30.8 vs 9.1%). (S2 Table). Our data show that among the 83 clients who had had an implant removed, 64.7% went to the Health center, 27.1% to the hospital and 8.2% in the resident of Nurse for removing the device (S3 Table).

## Discussion

This study found that 20% of Implanon acceptors had discontinued by 24 months post-insertion. This discontinuation rate is similar to the one found in a study conducted in Nigeria in which 19.3% had discontinued during the same period [18]. However, this discontinuation rate is lower to that found in the clinical trial on implant discontinuation, conducted in 21 centers in nine European countries, in which 31% had discontinued using the implant at 2 years [19]. The discontinuation rate in Kinshasa was also slightly lower than found in; Sudan

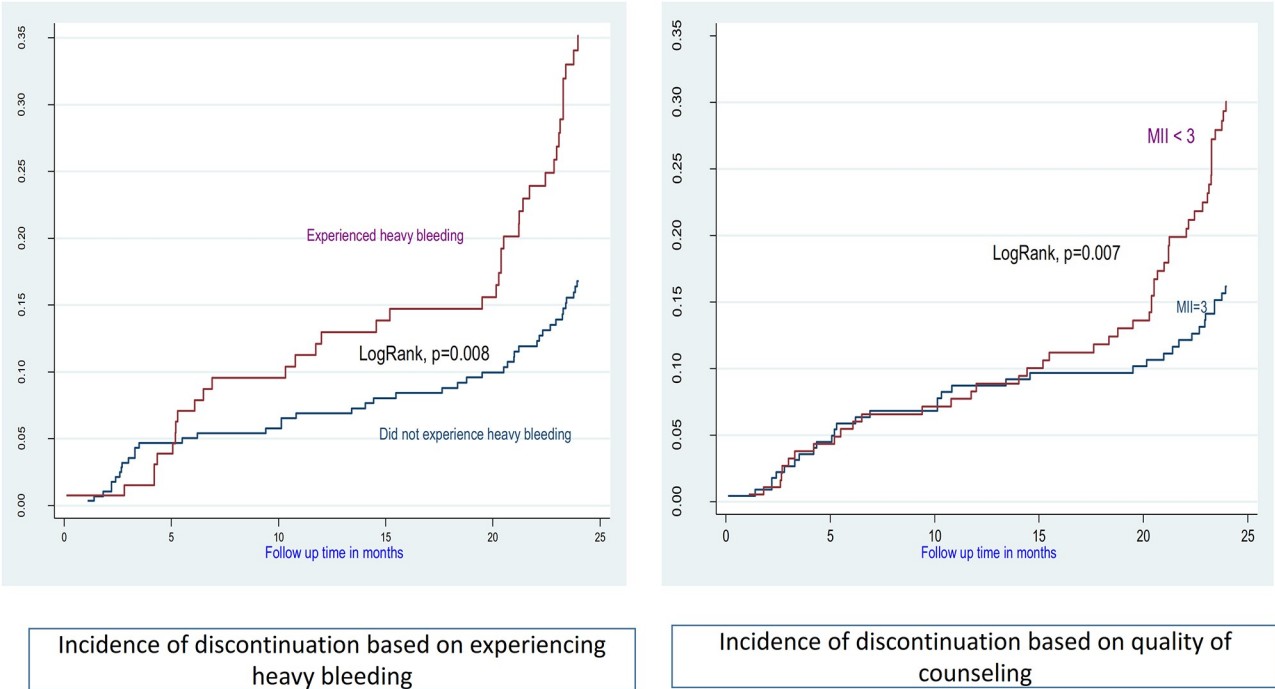

Fig 3. Incidence of discontinuation based on bleeding pattern and quality of counseling.

(43.5%) [20]; Ethiopia (46.5%), [21] and Australia (50%) [22], and another study from Nigeria [23]

In this analysis, predictors for early discontinuation included lower parity, residence in a military camp, lack of previous implant or injectable use, experience of heavy/prolonged bleeding, and low MII. Our findings on parity diverge from those in a study conducted in Ethiopia, in which for every additional child, the likelihood of modern contraceptive discontinuation decreased by 12% [24]. These multiparous women may be highly motivated not to remove Implanon early, since they are more likely to have achieved their desired family size and may have a higher level of motivation to avoid another pregnancy. Conversely, women with less children might be more likely to discontinue if they interpret the side effects as a risk to their future fertility. Controlling unwanted fertility with highly effective reversible contraception allowed couples to have the number of children they want at the time they want to have. However, fertility delay or impairment as a result of prior contraception use may lead to dissatisfaction and lower contraception use irrespective of actual desire [25, 26].

Women in military camps have limited access to modern family planning methods and are less likely to be using contraception [27]. In the DRC, military camps are closed environments and difficult to access by the civilian population; as such, they are not often targeted by mass health activities, including FP campaigns. Women living in military camps may have less access to a health facility, whereas those living near a health facility tend to have better knowledge of contraceptives than those living far from one [28, 29].

Dissatisfaction with bleeding patterns was the most common reason for early discontinuation, which is consistent with previous other studies [30, 31]. Counselling women on expected bleeding patterns has been shown to improve continuation rates for injectables and implant [30–36]. In this study, bleeding duration emerged as an important predictor of

**Table 2. Cox proportional hazard regression analysis of discontinuation rates.**

| Characteristics | total | Duration | Event | Incidence (1000 P-m) | Hazard ratio* | | | |
| | | | | | Crude (95%CI) | p | Adjusted (95%CI) | p |
|---|---|---|---|---|---|---|---|---|
| **Age** | | | | | | | | |
| • 18–24 | 146 | 3187.75 | 35 | 10.98 | 1.38 (0.89–2.14) | 0.146 | 0.75 (0.45–1.24) | 0.257 |
| • 25–49 | 269 | 5970.38 | 48 | 8.04 | 1 | | 1 | |
| **Instruction level** | | | | | | | | |
| • Less than secondary school | 81 | 1819.08 | 17 | 9.35 | 1.04 (0.61–1.77) | 0.897 | 0.98(0.55–1.76) | 0.955 |
| • Completed secondary school | 334 | 7339.05 | 66 | 8.99 | 1 | | 1 | |
| **Matrimonial Status** | | | | | | | | |
| • Not married | 339 | 7503.08 | 64 | 8.53 | 0.74(0.44–1.24) | 0.254 | 0.70 (0.41–1.22) | 0.214 |
| • Married/partnered | 76 | 1655.05 | 19 | 11.48 | 1 | | 1 | |
| **Residence** | | | | | | | | |
| • Military area | 94 | 1865.72 | 29 | 15.54 | 2.08 (1.33–3.27) | 0.001 | 2.28 (1.36–3.81) | 0.002 |
| • Non military area | 321 | 7292.41 | 54 | 7.40 | 1 | | 1 | |
| **SES** | | | | | | | | |
| • Lowest third | 126 | 2786.62 | 28 | 10.05 | 1.09 (0.65–1.83) | 0.732 | 1.48 (0.83–2.66) | 0.186 |
| • Middle third | 141 | 3122.62 | 25 | 8.01 | 0.86 (0.51–1.47) | 0.586 | 0.83 (0.48–1.44) | 0.509 |
| • Highest third | 148 | 3248.88 | 30 | 9.23 | 1 | | 1 | |
| **Number of living children** | | | | | | | | |
| • ≤ 2 | 164 | 3350.76 | 54 | 16.12 | 3.31 (2.11–5.20) | < 0.001 | 3.64(2.19–6.08) | < 0.001 |
| • 3 or more | 251 | 5807.38 | 29 | 4.99 | 1 | | 1 | |
| **Contraceptive history** | | | | | | | | |
| • Ever used injectable or implant | 45 | 1076.15 | 2 | 1.86 | 1 | | 1 | |
| • Never used injectable or implant | 370 | 8081.98 | 81 | 10.02 | 5.49 (1.35–22.32) | 0.017 | 4.49 (1.09–18.55) | 0.038 |
| **Experienced heavy/ prolonged bleeding** | | | | | | | | |
| • Yes | 131 | 2800.73 | 39 | 13.92 | 2.05 (1.33–3.16) | 0.001 | 1.96 (1.26–3.04) | 0.003 |
| • No | 284 | 6357.41 | 44 | 6.92 | 1 | | 1 | |
| **Method Information Index** | | | | | | | | |
| • < 3 | 188 | 4112.46 | 49 | 11.92 | 1.81 (1.17–2.81) | 0.008 | 2.08 (1.32–3.28) | 0.002 |
| • = 3 | 227 | 5045.67 | 34 | 6.74 | 1 | | 1 | |
| Overall | 415 | 9158.13 | 83 | 9.06 | | | | |

*p-m: person-months

discontinuation. This finding has several implications. Providers should recognize that medical advice on contraceptive side effects does not always resonate with women using the method; even light bleeding, when prolonged, may lead women to worry and discontinue use. Reported increases in length of bleeding predicted discontinuation, even when women's perceptions of those changes did not [34]. During counseling, providers should carefully discuss possible bleeding and actively encourage women to return in case of concern. During follow up visits by Implanon users, providers should examine the intensity and duration of vaginal bleeding to assess the gravity of the situation and rule out other causes such as infection, interactions with other medications, and gynecological pathology.

The quality of family planning counseling is a key issue in continuous use [37]. In this study low-quality family planning counseling, as measured by the MII score during the exit interview, was associated with subsequently higher 24-month discontinuation rates. This finding is consistent with the literature, which shows that a client's reception / understanding of

contraceptive information is a critical component in realizing full method choice and high-quality care [38]. This study confirms a consistent association of high MII scores and continuation rates on each round of follow up over the 24-month period [37].

Although other studies have found that quality of care is associated with subsequent continuation of contraceptive use [39–41], to the best of our knowledge, this is the first study conducted in a community setting to find a persistent pattern of high MII scores and lower rates of discontinuation over a 24-month follow-up period.

This study has several limitations. First is the possibility of misclassification of heavy bleeding, which women may have incorrectly reported. The bleeding should be recorded using the categories suggested by the WHO [42]. Second, there is a possibility of the response bias inherent to MII. Indeed, in the absence of observation of the counseling session, this analysis relies on self-reported data. Prior work comparing exit interview data to observation in facility surveys demonstrated that clients tended to overreport when asked if they received counseling about side effects of their method [43]. The recurring issue with the question: "were you told about side-effects?" is that it does not ask the more critical question: "and were you told that these side effects are normal and do not present a risk for your health / future fertility?" Although recall bias is also a potential limitation, the MII questions were asked to clients as part of an exit survey shortly after interacting with the provider.

The authors acknowledge that the performance of the nursing students could have been influenced by their awareness of being evaluated. Yet this influence could operate in opposing directions. Whereas they could perform better under observation than otherwise, they might also performance less well from being nervous about the observation. Another potential limitation of this study is that the research team could not ascertain discontinuation rates of the women who were LTFU.

Finally, although we have used Cox regression, we cannot exclude a potential confounding effect of other variables such as the effect of women's autonomy on method discontinuation.

To our knowledge, this study is the first to report continuation rate for implants offered at the community level (outside a fixed clinic). The study reinforces the importance of the quality of FP counselling in method continuation. Furthermore, it shows that dissatisfaction with bleeding patterns can push women to remove Implanon early. These findings provide useful programmatic guidance to policy makers, healthcare workers, and other stakeholders involved in FP programming in helping to understand factors associated with discontinuation of Implanon NXT at the community level.

## Conclusions

In addition to four client characteristics that predicted discontinuation, we identified one programmatic factor: quality of counseling as measured by the Method Information Index. Community providers in similar contexts should pay more attention to clients having less than three children, new adopters, and to clients living military camps as underserved population, where clients have less access to health facilities. More targeted counselling and follow-up is needed, especially on bleeding patterns.

## Supporting information

**S1 Table. Cumulative discontinuation rate over time.**
(DOCX)

**S2 Table. Reasons for early discontinuation stratified by predictors.**
(DOCX)

**S3 Table. Site where the clients went to remove the device.**
(DOCX)

## Author Contributions

**Conceptualization:** P. Z. Akilimali, Hernandez J., Kayembe K. P., Bertrand J.

**Data curation:** P. Z. Akilimali.

**Formal analysis:** P. Z. Akilimali, Bertrand J.

**Funding acquisition:** P. Z. Akilimali, Bertrand J.

**Investigation:** P. Z. Akilimali, Bertrand J.

**Methodology:** P. Z. Akilimali, Hernandez J., Anglewicz P., Kayembe K. P., Bertrand J.

**Project administration:** P. Z. Akilimali, Bertrand J.

**Resources:** P. Z. Akilimali, Bertrand J.

**Software:** P. Z. Akilimali.

**Supervision:** P. Z. Akilimali, Hernandez J., Bertrand J.

**Validation:** P. Z. Akilimali, Kayembe K. P., Bertrand J.

**Visualization:** P. Z. Akilimali, Bertrand J.

**Writing – original draft:** P. Z. Akilimali, Hernandez J., Anglewicz P., Kayembe K. P., Bertrand J.

**Writing – review & editing:** P. Z. Akilimali, Hernandez J., Anglewicz P., Kayembe K. P., Bertrand J.

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
