## [Decision Letter · Decision Letter 0]

21 Feb 2020

PONE-D-19-25274

Incidence and Determinants of Implanon Discontinuation: Findings from a Prospective Cohort Study in Three Health Zones in Kinshasa, DRC

PLOS ONE

Dear Dr Akilimali,

Thank you for submitting your manuscript to PLOS ONE. After careful consideration, we feel that it has merit but does not fully meet PLOS ONE’s publication criteria as it currently stands. Therefore, we invite you to submit a revised version of the manuscript that addresses the points raised during the review process.

We would appreciate receiving your revised manuscript by Apr 06 2020 11:59PM. To enhance the reproducibility of your results, we recommend that if applicable you deposit your laboratory protocols in protocols.io, where a protocol can be assigned its own identifier (DOI) such that it can be cited independently in the future. For instructions see: http://journals.plos.org/plosone/s/submission-guidelines#loc-laboratory-protocols

We look forward to receiving your revised manuscript.

Kind regards,

Catherine S. Todd

Academic Editor

PLOS ONE

Journal Requirements:

2. Please include additional information regarding the survey or questionnaire used in the study and ensure that you have provided sufficient details that others could replicate the analyses. For instance, if you developed a questionnaire as part of this study and it is not under a copyright more restrictive than CC-BY, please include a copy, in both the original language and English, as Supporting Information. Also, pre-testing of the questionnaire was mentioned but not on whom this pre-testing took place and how many participants were involved. Please update with this information as appropriate.

Additional Editor Comments (if provided):

This is a well-written manuscript regarding efforts to broaden access to long-acting reversible contraception in a context with a very low contraceptive prevalence rate. I agree with the reviewer that further information regarding context, sample selection, and results will enhance the manuscript and will be pleased to see a revised submission.

Reviewers' comments:

Reviewer's Responses to Questions

**Comments to the Author**

1. Is the manuscript technically sound, and do the data support the conclusions?

Reviewer #1: Yes

2. Has the statistical analysis been performed appropriately and rigorously? 

Reviewer #1: I Don't Know

3. Have the authors made all data underlying the findings in their manuscript fully available?

Reviewer #1: Yes

4. Is the manuscript presented in an intelligible fashion and written in standard English?

Reviewer #1: Yes

5. Review Comments to the Author

Reviewer #1: Methodology: page 4 -perhaps use another word instead of "opted" rather use chose? to use Implanon.

Can you comment on the experience of nursing students to provide the service. Did they provide a better service as they knew research would evaluate their service provision.

What is meant by providing services at community level outreach event? was it a campaign to specifically target contraceptive uptake or the Implant? what other methods were available ? other helath services?Could they have got implanon at a local service nearby (for post insertion support)

You say it was not possible to interview all 702, so what was the convenience sample limited to? those who agreed to stay?

Why were the three districts chosen? noting one was very different? was there a programme that wanted to ensure IMplanaon wa smade available to the women in these areas?

Did you collect data on who removed the implant? was it the same nurses and where did they go?

The methodology says women were interviewed on the phone? on page 5 but a few sentance later it says women were asked to appear ? at a central community location. these two statements contradict each other. If many women lived in a military camp why did the study not ask for permission to interview them at the camp or did they also have to go to a community venue? again in analytic methods it says data was collected on a mobile phone?

Did women get reimbursed for participation

Results:

Could you have tracked the LTFP via clinics in case they had the implant removed?

Conclusions: The woman in military camps - were they working or just living with men employed by the military? if working perhaps the heavy bleeding impacted on their work or the camps had poor facilities? if they lack access to health facilities how did they get their implants removed? and if they all received equal conselling on a variety of methods why would they have less knowledge than other woman. Is it that the camps provide no Health facilities to their staff?

Later in the conclusion it says the study was conducted in a community setting? rather enrollment was conducted in a community setting but interviews were on the phone? not in the community ? see earlier comments.

6. PLOS authors have the option to publish the peer review history of their article (what does this mean?). If published, this will include your full peer review and any attached files.

Reviewer #1: No

---

## [Author Response · Author response to Decision Letter 0]

15 Mar 2020

Comments from the reviewer are below in bold font, and our responses follow each comment in plain font

Reviewer #1: 

Page 4 -perhaps use another word instead of "opted" rather use chose? to use Implanon.

We agree and have updated accordingly.

Can you comment on the experience of nursing students to provide the service. Did they provide a better service as they knew research would evaluate their service provision. 

We appreciate this comment. To address this issue, we have added the following text to the discussion section: “The authors acknowledge that the performance of the nursing students could have been influenced by their awareness of being evaluated. Yet this influence could operate in opposing directions. Whereas they could perform better under observation than otherwise, they might also performance less well from being nervous about the observation.”

What is meant by providing services at community level outreach event? was it a campaign to specifically target contraceptive uptake or the Implant? what other methods were available ? other health services? Could they have got implanon at a local service nearby (for post insertion support)

As part of their practical training, the students provided FP services at the community level: at outreach events (mini-campaigns) set up in a central location (e.g., near a referral health facility, beside a market), door-to-door, or from their own homes. Nursing students offered a range of five contraceptive methods: Implanon NXT, DMPA-SC, oral pills, condoms, and Cycle beads. Prospective clients often preferred these community services to fixed facilities in their neighborhoods, because they were less expensive and required less waiting time. 

You say it was not possible to interview all 702, so what was the convenience sample limited to? those who agreed to stay?

In addition to “willingness to stay,” the respondent needed to provide consent for the interview. So, out of the 702 women who received Implanon NXT, 531 women aged 18 to 49 years old agreed to participate in the study and gave their informed consent to be interviewed at 6,12 and 24 months.

Why were the three districts chosen? noting one was very different? was there a programme that wanted to ensure Implanon was made available to the women in these areas?

The selection of zones took into account the proximity of the training institutes in which the students were trained. In addition, for logistical efficiency, we needed to limit that number of health zones to be covered and selected three with different characteristics to capture different types of clients (to increase the generalizability of the results). 

Did you collect data on who removed the implant? was it the same nurses and where did they go? 

Nursing students received training in the removal of Implanon NXT in the 4th year of nursing school. At the large majority of outreach events, a more experienced nurse handled the removal of implants (not the nursing students), and often the client was referred to the health facility near the outreach site for the removal. Our data show that among the 83 clients who had had an implant removed, 65% went to the Health center, 27% to the hospital and 8% in the resident of Nurse to remove the device. 

The methodology says women were interviewed on the phone? on page 5 but a few sentences later it says women were asked to appear ? at a central community location. these two statements contradict each other. If many women lived in a military camp why did the study not ask for permission to interview them at the camp or did they also have to go to a community venue? again in analytic methods it says data was collected on a mobile phone?

During this study, trained interviewers used smart phone technology to collect data during a face-to-face interview. The use of mobile technology for data collection allowed interviewers to automatically upload data to a secure electronic server instead of having to code and enter data manually. Data was collected using interview face to face using mobile phone as tools ( SurveyCTO). These were not phone interviews. However women were re-contacted by phone to schedule the follow-up interviews.

Did women get reimbursed for participation?

Participants received U.S. $5.00 as reimbursement for transportation costs.

Could you have tracked the LTFP via clinics in case they had the implant removed?

Unfortunately, this is not possible. The client record system at these clinics would not have this type of information, nor is it clear that women using the community service would ever have registered with the nearby clinic. 

The woman in military camps - were they working or just living with men employed by the military? if working perhaps the heavy bleeding impacted on their work or the camps had poor facilities? if they lack access to health facilities how did they get their implants removed? and if they all received equal counselling on a variety of methods why would they have less knowledge than other woman. Is it that the camps provide no Health facilities to their staff?

Most of the participants from Military camps were the wives of military personnel. Military camps are closed settings in which the health structures are less likely to receive technical and financial support from development partners. Also, results from a previous survey show that women in military camps have lower levels of knowledge and use of contraception than do their civilian counterparts. Because one of these development partners has begun supporting the military camp health facilities in recent years, Implant users have access to trained personnel who can remove implants. 

Later in the conclusion it says the study was conducted in a community setting? rather enrollment was conducted in a community setting but interviews were on the phone? not in the community? see earlier comments.

The study was conducted in a community setting and participants were interviewed face to face using a questionnaire programmed on a mobile phone. During this study, trained interviewers used smart phone technology to collect data. The use of mobile technology for data collection allowed interviewers to automatically upload data to a secure electronic server instead of having to code and enter data manually. Data was collected during face to face interviews using mobile phone as tools ( SurveyCTO). This was not a phone survey.

---

## [Editor Report · Decision Letter 1]

10 Apr 2020

PONE-D-19-25274R1

Incidence and Determinants of Implanon Discontinuation: Findings from a Prospective Cohort Study in Three Health Zones in Kinshasa, DRC

PLOS ONE

Dear Dr Akilimali,

Thank you for submitting your manuscript to PLOS ONE. After careful consideration, we feel that it has merit but does not fully meet PLOS ONE’s publication criteria as it currently stands. Therefore, we invite you to submit a revised version of the manuscript that addresses the points raised during the review process.

We would appreciate receiving your revised manuscript by May 25 2020 11:59PM. To enhance the reproducibility of your results, we recommend that if applicable you deposit your laboratory protocols in protocols.io, where a protocol can be assigned its own identifier (DOI) such that it can be cited independently in the future. For instructions see: http://journals.plos.org/plosone/s/submission-guidelines#loc-laboratory-protocols

We look forward to receiving your revised manuscript.

Kind regards,

Catherine S. Todd

Academic Editor

PLOS ONE

Additional Editor Comments (if provided):

This manuscript has largely responded to the queries raised by the prior reviewer. It will be suitable for publication once two minor changes are made:

- Please add the description of the participants' context as military wives to the manuscript, ideally in the Introduction section. You identify the program as novel but do not mention the circumstances of the context where family planning services are traditionally limited and women less aware of contraceptive options.

- Please add the reimbursement of travel expenses to the Methods section - this is a required feature to report.

---

## [Author Response · Author response to Decision Letter 1]

13 Apr 2020

April 14th, 2020

To: Catherine S. Todd

Academic Editor

PLOS ONE

Re: Revised version manuscript PONE-D-19-25274 R2 

Dear Editor,

Thank you very much for the opportunity to revise and resubmit our manuscript. The comments we received helped to improve our research. 

Our revised submission includes the following documents: (1) the manuscript with changes tracked, (2) a clean revised manuscript including tables and (3) the rebuttal letter with our responses to each point raised by the editor and reviewers. 

Below is our reply to the comments and suggestions of the reviewers on a point-by-point basis.

We would like to thank the referees for the valuable and constructive comments and suggestions. We are hoping that we have adequately dealt with the comments and suggestions, and that the revised version is to your satisfaction. 

Looking forward to hearing from you.

Sincerely,

Dr. Pierre Zalagile Akilimali

Kinshasa School of Public Health

University of Kinshasa

Democratic Republic of Congo 

 

Reviewer’s Comments

Please add the description of the participants' context as military wives to the manuscript, ideally in the Introduction section. 

Kokolo health zone consists of compounds that house military personnel and their families. Military compounds differs from other health zones in that they are a closed encampment within the city with their own health system, though often underfunded and affording less access to family planning than in the non-military population. We have added a statement about this on page 5 in Method section.

You identify the program as novel but do not mention the circumstances of the context where family planning services are traditionally limited and women less aware of contraceptive options.

Please add the reimbursement of travel expenses to the Methods section - this is a required feature to report.

We have added the following statement “Participants received U.S. $5.00 as reimbursement for transportation costs per visit.” to page 7 of the manuscript.

---

## [Editor Report · Decision Letter 2]

20 Apr 2020

Incidence and Determinants of Implanon Discontinuation: Findings from a Prospective Cohort Study in Three Health Zones in Kinshasa, DRC

PONE-D-19-25274R2

Dear Dr. Akilimali,

We are pleased to inform you that your manuscript has been judged scientifically suitable for publication and will be formally accepted for publication once it complies with all outstanding technical requirements.

With kind regards,

Catherine S. Todd

Academic Editor

PLOS ONE
---

## [Editor Report · Acceptance letter]

28 Apr 2020

PONE-D-19-25274R2 

Incidence and Determinants of Implanon Discontinuation: Findings from a Prospective Cohort Study in Three Health Zones in Kinshasa, DRC 

Dear Dr. Akilimali:

I am pleased to inform you that your manuscript has been deemed suitable for publication in PLOS ONE. Congratulations! Your manuscript is now with our production department. 

With kind regards,

on behalf of

Dr. Catherine S. Todd 

Academic Editor

PLOS ONE